# Exploring the cytotoxic mechanisms of Pediocin PA-1 towards HeLa and HT29 cells by comparison to known bacteriocins: Microcin E492, enterocin heterodimer and Divercin V41

George P. Buss[ID]*, Cornelia M. Wilson

School of Life Sciences, Canterbury Christ Church University, Canterbury, United Kingdom

* george.buss97@hotmail.com

## Abstract

The purpose of this study was to explore potential mechanisms of cytotoxicity towards HeLa and HT29 cells displayed by Pediocin PA-1. We did this by carrying out sequence alignments and 3D modelling of related bacteriocins which have been studied in greater detail: Microcin E492, Enterocin AB heterodimer and Divercin V41. Microcin E492 interacts with Toll-Like Receptor 4 in order to activate an apoptosis reaction, sequence alignment showed a high homology between Pediocin PA-1 and Microcin E492 whereas 3D modelling showed Pediocin PA-1 interacting with TLR-4 in a way reminiscent of Microcin E492. Furthermore, Pediocin PA-1 had the highest homology with the Enterocin heterodimer, particularly chain A; Enterocin has also shown to cause an apoptotic response in cancer cells. Based on this we are led to strongly believe Pediocin PA-1 interacts with TLRs in order to cause cell death. If this is the case, it would explain the difference in cytotoxicity towards HeLa over HT29 cells, due to difference in expression of particular TLRs. Overall, we believe Pediocin PA-1 exhibits a dual effect which is dose dependant, like that of Microcin. Unfortunately, due to the COVID-19 pandemic, we were unable to carry out experiments in the lab, and the unavailability of important data meant we were unable to provide and validate out solid conclusions, but rather suggestions. However, bioinformatic analysis is still able to provide information regarding structure and sequence analysis to draw plausible and evidence based conclusions. We have been able to highlight interesting findings and how these could be translated into future research and therapeutics in order to improve the quality of treatment and life of cancer patients.

## Introduction

From 2015–2017 there were around 367,000 people given a new diagnosis of cancer every year, with breast, prostate, lung and bowel cancer accounting for 53% of these new diagnoses [1]. This study investigated the effect of Pediocin PA-1 on HT-29 cells, a cell line isolated in 1964 from colonic adenocarcinoma cells [2]; and HeLa cells, a cell line isolated in 1951 from cervical cancer cells [3]. Considering the neurotoxic effects of conventional therapies, such as

**Data Availability Statement:** All relevant data are within the manuscript.

**Funding:** The authors received no specific funding for this work.

**Competing interests:** The authors have declared that no competing interests exist.

chemotherapy and radiotherapy, from minor cognitive effects to major pathology such as encephalopathy [4, 5], the exploration of bacteriocins as a novel anti-cancer therapy allows the opportunity for a better quality of life for cancer patients undergoing treatment. Further still, targeted therapies have shown to improve the longevity and quality of patients lives [6, 7]. Bacteriocins offer the opportunity for the development of highly targeted therapies whilst still ensuring an even greater quality of life.

Pediocin PA1 is a 62-amino acid long class IIa bacteriocin expressed in *Pediococcus acidilactiti* (gram-positive bacteria) generally in response to stress and/or ultraviolet light [8, 9]. Bacteriocins are catatonic peptides produced by all types of bacteria that are non- immunogenic, biodegradable and can colonise cancer cells with specific toxicity [10]. Pediocin has been shown to display cytotoxic effects towards HeLa and HT29 cells, with a greater cytotoxic effect towards HeLa over HT29 [11]. Whilst there have been several studies discussing the cytotoxic effect of Pediocin, the mechanism has never been studied in as great detail as other bacteriocins. This is a comparative study against other bacteriocins which have been previously thoroughly researched; it is hoped that by carrying out sequence alignments and 3D modelling we will be able to identify potential mechanisms of actions by Pediocin PA-1. Microcin E492 and Enterocin AB heterodimer have both been shown to induce apoptosis, indicating a protein interaction [12–14]. Therefore by comparing sequence alignment and analysing 3D models we hope to identify similarities within the structure of Pediocin A1 compared to these bacteriocins which may give further insight into its mechanism of action.

Furthermore, Divercin V41 is also a class IIa bacteriocin which was shown to have no cytotoxic effect against HT29, despite it belonging to the same class of toxin as Pediocin PA-1, we hope to identify the differences between these two bacteriocins, and thereby gain greater insight into Pediocin PA-1's mechanism of action.

There has been some controversy pertaining to the mechanism of Pediocin PA-1: four cysteine residues within the structure indicated disulphide bridge formation, which forms a poration complex in target membrane, leading to cell death [15]. It has also been reported that Pediocin PA-1 is able to function in the absence of protein receptors [16]. However, Enterocin has also shown to cause the lipid bilayer permeabilization whilst inducing Apoptosis as well by bio-energetic collapse [17]–this is indicative of protein interaction with Toll-Like Receptors (TLRs) to trigger a caspase response. This would suggest a dual mechanism of action, as Enterocin is able to cause cell death through separate mechanisms of permeabilization of the lipid bilayer, leading to cell lysis, as well as inducing apoptosis by bio-energetic collapse through probable TLR interaction.

Other bacteriocins have also demonstrated to have this dual mechanism of cytotoxicity–which we discuss later in this paper. Therefore, we argue that it is likely Pediocin PA-1 also exhibited a dual mechanism of cytotoxicity due to previous observations, as well as similarity in shape and sequence with other bacteriocins with dual mechanisms of cytotoxicity–all of which are discussed later in this paper.

Whilst bacteriocins as a therapeutic agent have not proceeded past animal studies, the results of these studies clearly demonstrate how bacteriocins offer a potential advancement in the treatment of cancer, as most target cancer cells whilst having very limited interaction with human cells. This is due to the negative charge of cancer cells [18], the negative charge has been labelled the "Warburg Effect" which explains the secretion of more than thirty times the amount of lactic acid than healthy cells, by cancer cells [19, 20]. Bacteriocins have an overall positive charge, and thus target cancer cells over human cells [10]. Pediocin PA-1 itself is non-toxic, nonimmunogenic being used as a bio-food preservative protecting against *Listeria monocytogenes* [21], other bacteriocins are also widely used in the same manner. In this way we know that Pediocin is safe for human consumption.

Whilst this study had its limitations due to the COVID-19 pandemic preventing us from carrying out *in vitro* experiments in the lab, we were still able to utilise bioinformatic tools to gain a further insight into the potential mechanisms of Pediocin PA-1. Whilst also commenting on other bacteriocins. In this way we were able to provide a fundamental understanding at a molecular level for further investigation, as well as highlighting the potential use of bacteriocins as novel cancer therapeutics.

## Methodology

### Accessing protein sequences

UniProt [22] was used to access the protein sequences in this analysis (Table 1). UniProt is an opensource database maintained by the UniProt consortium. The database is an amalgamation of Swiss-Prot, TrEMBL and the PIR Protein Sequence Database. It contains protein sequence and functional information often derived from primary genome research and analysis.

Pediocin PA1 and the other bacteriocins were located on the UniProt database, the PBD 3D structure was then downloaded. In the case of Divercin V41 there was no current PBD model, so the protein sequence was downloaded and modelled using Swiss Model [23–25].

### Sequence alignment

Protein sequences were downloaded from UniProt and then aligned using MultiAlin [22, 26]. Pediocin, Divercin, Microcin and Enterocin were uploaded to the server, default parameters of 90% high consensus and 50% low consensus were used. The sequence alignment was exported as a table image.

### Model analysis

Once downloaded, PDB files were viewed in VMD [27]. VMD is an opensource modelling software which allows visualisation and analysis of protein structures. VMD can also be used to simulate and analyse the molecular dynamics of a system. We were able to use VMD to identify interacting residues that appeared significant to each protein structure and propose a mechanism of action based on this and previous *in-vitro* findings. It is worth noting that in the case of TLR-4 we located an accurate model of the LPS Ra complex of E. coli and TLR-4 interacting from RCSB PDB (named 3FXI). We then isolated the TLR-4 using the viewing tool of RCSB PDB and downloaded the.pdb file [28].

### Swiss-model

Divercin V41 did not have a published PBD file on UniProt and we were unable to identify any previous research which had attempted to deduce the 3D structure. Therefore, we used Swiss-Model to derive a predicted structure for Divercin V41, this was downloaded as a PDB

**Table 1. Table showing the proteins analysed and their corresponding accession numbers as according to UniProt [22].**

| Protein | Acession_Number |
| --- | --- |
| Pediocin PA1 | P29430 |
| Divercin V41 | Q9Z4J1 |
| Microcin E492 | A0A652PYJ5 |
| Enterocin A | AF240561 |
| Enterocin B | AYG20277 |

```
                 1        10        20        30        40        50        60        70        80        90       100 102
                 |--------+---------+---------+---------+---------+---------+---------+---------+---------+---------+-|
Toll-like                                          MNITSQMNKTIIGVSVLSVLVYSVVAVLVYKFYFHLMLLAGCIKYGRG
Bacteriocin     MKKIEKLTEKEMANIIG-GKYYGNGVTCGKHSCSVDWGKATTCIINNGAMAWATGGHQGNHKC
Microcin        MREISQKDLNLAFGAGETDPNTQLLNDLGNNMAWGAALGAPGGLGSAALGRAGGALQTVGQGLIDHGPVNVPIPVLIGPSWNGSGSGYNSATSSSGSGS
Consensus       .........k....g.g.....n.........g..a.g..g..a.g..g.....v...................l.g....g.g............
```

**Fig 1. Sequence comparison of Microcin E492 and Pediocin A1 (pediocin).** Low consensus alignments (50%) are represented as blue letters, whilst high consensus alignments (90%) are represented as red letters. Microcin is significantly larger than Pediocin A1, whilst there is homology this a mainly low consensus homology with fourteen residues of high consensus [26].

file and viewed in VMD. We were also unable to locate a fully devised 3D structure for Enterocin B, so Swiss-model was also used in this instance. SWISS-MODEL is an automated protein structure homology-modelling server [23–25, 29].

# Results

## Results Microcin E492

Microcin E492 is a highly hydrophobic 7.9kDa bacteriocin produced by *Klebsiella pneumonae* [30, 31]. When co-cultured with HeLa cells cytotoxic effects observed were typical of apoptosis, these included: cell shrinkage, DNA fragmentation, caspase-3 activation and loss of mitochondrial membrane potential; cell necrosis was also observed at higher doses of Microcin [13]. Additionally, the presence of zZAD-fmk (caspase inhibitor) completely blocked the cytotoxic effect of Microcin [13]. Caspase 3- activation is associated with the activation of toll-like receptor 4, therefore during the 3D modelling analysis we included toll-like receptor 4 (TLR-4) to observe how Microcin E492 interacts with it [32].

**Sequence alignment.** A sequence alignment was performed between Microcin E492 and Pediocin PA-1. Microcin E492 is a significantly larger protein than Pediocin, however despite this difference there is significant homology between the two proteins (See Fig 1). Largely, the homology is of low consensus with only fourteen residues aligning with a high consensus.

**3D modelling: VMD.** Unfortunately, despite extensive research there was no 3D model available for Microcin E492, therefore we used SWISS-MODEL to build a 3D model based on the fasta sequence obtained from RCSB PDB [33]. This was then downloaded as a.pdb file and viewed in VMD alongside the toll-like receptor-4. The model built using SWISS-MODEL had a sequence identity of 30%, whilst this is not ideal it was the only best possible model of Microcin E492 available at the time of research.

TLR-4 is arranged as a heterodimer of two chains: A and B, they are arranged in a helical structure and the ends of each chain come together to fold around each other. According to VMD analysis (see Fig 2), Microcin E492 sits in the middle of the heterodimer where both

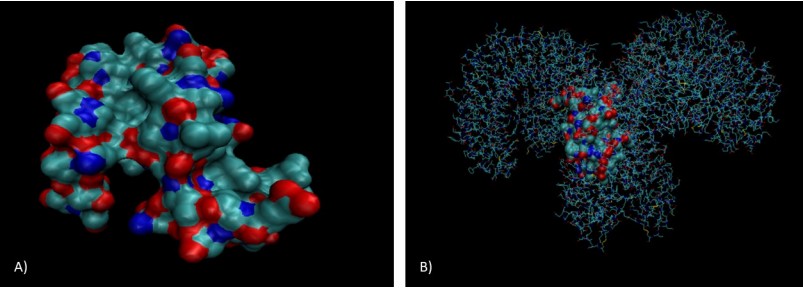

**Fig 2.** 3D model of microcin (A) and microcin interacting with TLR-4 (B) produced using the.pdb file imported from RCSB PDB to VMD. TLR-4 has been drawn in lines in order to distinguish between microcin and TLR-4 A) Microcin appears mainly globular in shape and has small yet frequent regions of polarity. B) TLR-4 is arranged as a heterodimer of two chains converging in the centre, microcin interacts with both chains in the middle of this convergence [27].

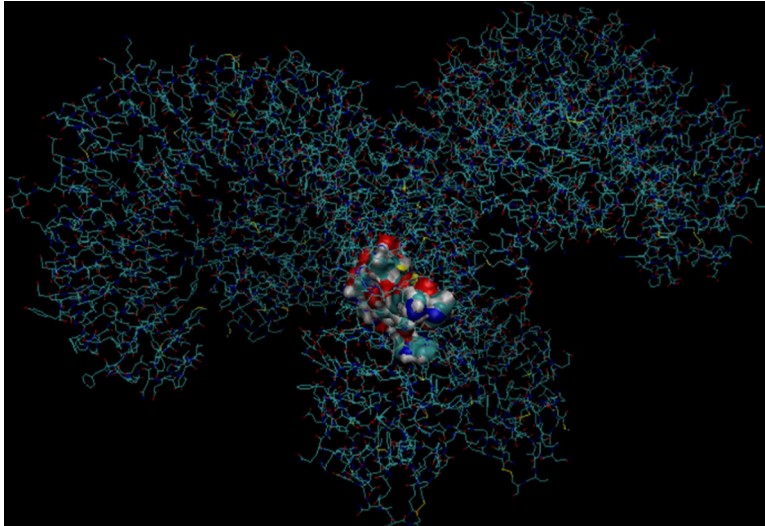

**Fig 3. 3D model of Pediocin A1 interacting with TLR-4 produced using the.pdb file imported from RCSB PDB to VMD.** TLR-4 has been drawn in lines in order to distinguish between Pediocin A1 and TLR-4. Like microcin, pediocin appears mainly globular in shape, the TLR-4 is arranged as a heterodimer of two chains converging in the centre, Pediocin A1 interacts with both chains in the middle of this convergence [27].

chains meet, interacting with both chains. Common contact residues for both chain A and B with Microcin included mainly hydrophobic residues: glycine, valine, phenylalanine, leucine and, isoleucine.

In order to explore the possibility that Pediocin PA-1 also could interact with TLR-4 we carried out 3D modelling of Pediocin PA-1 interacting with TLR-4 too (see Fig 3). Whilst being smaller than Microcin E491, Pediocin also sat in between the receptor interacting with both chains as they met. Common contact residues were the same as Microcin E492: glycine, valine, phenylalanine, leucine and isoleucine.

## Enterocin

Enterocin is a class IIa bacteriocin, it is a heterodimer formed from chains A and B and is produced by *Enterococcus faecium*. It is used within different food products due to its anti-listerial properties [14, 17]. The heterodimer is formed by strong hydrophobic forces of leucine, isoleucine, tyrosine, glycine and phenylalanine residues [34]. Enterocin also displays cytotoxic effects towards HT29 and HeLa cells, as a heterodimer these effects were greater. Enterocin also displayed a greater effect towards HeLa than HT29 cells [14, 35].

Sequence alignment and 3D models of Enterocin A and B were completed in order to distinguish whether there were any shared features between the mechanism of action of Pediocin and the Enterocin heterodimer.

**Sequence alignment.** A sequence alignment was carried out for PediocinA1, Enterocin A and, Enterocin B. The previous research has highlighted potentially have different mechanisms of action for Enterocin A and B, therefore upon comparing Pediocin PA-1 to each homodimer it can provide further insight into the exact mechanism of action of Pediocin PA-1. The highest homology is seen between Pediocin and Enterocin chain A, however there is still a significant homology between Pediocin and Enterocin B (see Fig 4). This could indicate that Pediocin has a dual mechanism of action incorporating how Enterocin functions as a heterodimer.

```
        1        10        20        30        40        50        60      7072
        |--------+---------+---------+---------+---------+---------+---------+-|
sp|P29430|Pediocin  MKKIEKLTEKEMANIIGG-----K----YYGNGVTCGKHSCSVDWGKATTCIINNGAMAWATGGHQGNHKC
tr|Q9L658|EntA      MKHLKILSIKQTQLIYGGTTHSGK----YYGNGVYCTKNKCTVDWAKATTCIAGMSIGGFLGGAIPG--KC
tr|O34017|EntB      MQNVKELSTKEMKQIIGGENDHRMPNELNRPNNLSKGGAKCGARIAGGLFGIP-KGPLAWRAGLANVYSKCN
Consensus           Mk..k.Ls.K#m..IiGG.....k....yygNgv.cgk.kC.vdwakattcI...g..awa.G...g..KC.
```

**Fig 4. Sequence comparison of Enterocin A (EntA), Enterocin B (EntB) and Pediocin A1 (pediocin).** Low consensus alignments (50%) are represented as blue letters, whilst high consensus alignments (90%) are represented as red letters. Pediocin has a high level of alignment with both Enterocin A and Enterocin B [26].

**3D modelling analysis.** Enterocin A is significantly smaller than Enterocin B, 712 residues compared to 2125 residues. Enterocin B also appears to have a greater charged surface than Enterocin B. It is also interesting to note that both molecules have two cysteine residues (shown in yellow) on their surface (see Figs 5 and 6), however when acting as a homodimer these residues do not seem to be involved and no disulphide bridges were detected on analysis. Further analysis revealed a hydrophobic surface of Enterocin, with exposed residues including: Val'15, Trp'33, Lys'43. Tyr'2, Trp'33 and Ala'32. This is in keeping with the previous findings [14].

## Divercin V41

Divercin has previously been shown to have a similar homology to Pediocin whilst showing no anti-tumour properties towards HT29 cells [36]. Therefore, we performed sequence alignment and 3D modelling of Divercin V41 in order to try and understand why, despite the similar homology, one shows anti-tumour effects towards HT29 cells and not the other.

**Sequence alignment.** The sequence comparison of Divercin V41 and Pediocin PA-1 shows a high homology, with high consensus (90% consensus) proteins including hydrophobic residues glycine and isoleucine; amphipathic residues tyrosine and tryptophan; and charged residues of glutamic acid (see Fig 7). Charged residues of glutamic acid are often used in the formation of salt bridges. Low consensus proteins (50% consensus) include: hydrophobic residues glycine, isoleucine, alanine; charged residues lysine, aspartic acid, and lysine; polar charged residues histidine and glutamine. Polar charged residues are often associated with hydrogen bond formation through acting as protein donators and acceptors.

**3D modelling analysis.** The devised 3D model from SWISS MODEL (see Fig 8) predicted Divercin V41 to be more linear in shape compared to Pediocin PA-1, which is more globular. Pediocin PA-1 is also shown to have a greater surface polarity than Divercin V41. Depending

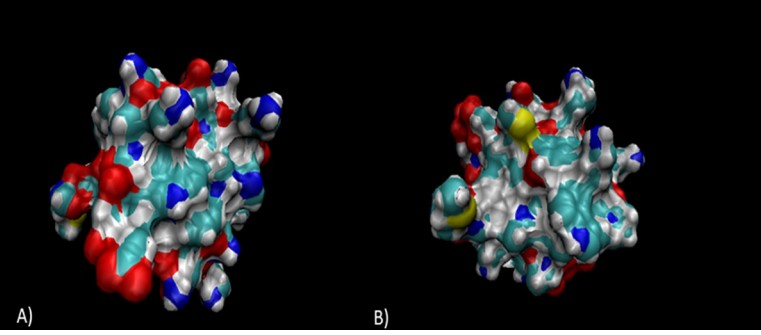

**Fig 5. 3D model of Enterocin A produced using the.pdb file imported from RCSB PDB to VMD.** A) shows the positive x/y axis angle of Enterocin A, whilst B) shows the negative x/y angle. Analysis reveals regions of polar and charged residues (shown in red) of the surface of Enterocin B, as well as two cysteine residues (shown in yellow) on the surface of Enterocin A [27].

**Fig 6.** Devised 3D model of Enterocin B alongside the global quality estimate (C). Analysis reveals areas of polarity (A) and one cysteine residue (B). The overall sequence identity was 7.69% however, there is currently no 3D model of Enterocin B on it's own, therefore this is the most accurate representation to date [23–25].

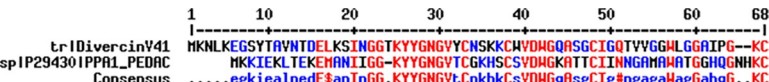

**Fig 7. Sequence comparison of Divercin V41 and Pediocin A1 (shown as PPA1_PEDAC).** Low consensus alignments (50%) are represented as blue letters, whilst high consensus alignments (90%) are represented as red letters. The overall consensus sequence is low alignment [26].

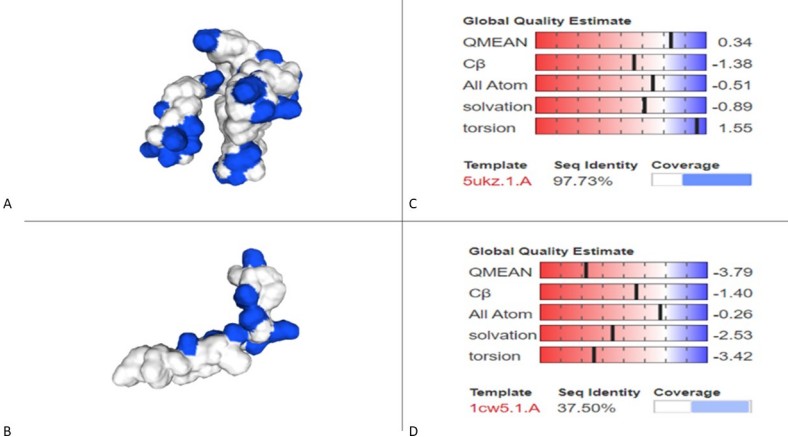

**Fig 8.** Devised 3D models of Divercin V41 (B) and Pediocin A1 (A) alongside their global quality estimate respectively (C and D). Pediocin is shown to have a greater surface polarity than Divercin. There is only a 37.50% sequence identity for the DIvercin V41 model compared to the Pediocin A1 model, however due to the lack of data this is the most accurate model that we are aware of [23–25].

on the mechanism of action, both the difference in shape and polarity could explain up to a certain extent, the reason for Divercin V41's inability to show anti-tumour effects on HT29 cells despite the relatively high homology with Pediocin PA-1.

VMD analysis revealed exposed residues included charged residues: Asp'41, Lys'25, Lys'36 and Lys'65; polar residues: Asn'29, Asn'34, Gln'44, Gln'51, Tyr'2 and Tyr'32; hydrophobic residues: Gly'5, Gly'39 and Gly'64; as well as amphipathic Trp'42. There were also three Cysteine residues on the -y axis of the Divercin V41. Despite this, there were no salt bridges detected upon investigation.

## Discussion

Like Pediocin PA-1, Microcin E492 has shown cytotoxic effects against HeLa cells. The observations of this cytotoxicity are inductive of apoptosis both biochemically and morphologically [13]. Further analysis revealed this apoptosis was due to caspase 1 and 3 activation. Definitive support can be found in the same study, as the use of zZad-fmk (a general caspase inhibitor) inhibited the cytotoxic effect of Microcin. Caspase 3 activation has been linked to the activation of TLR-4 [36], therefore we are confident in our conclusion that Microcin must bind to TLR-4. Caspase 1 activation has been linked to the efflux of potassium ions, which activates the NLRC4 (nucleotide-binding oligomerization domain and leucine-rich repeat containing receptors) inflammasome pathway leading to the assembly of caspase 1 [37]. Considering previous studies have found high doses of Microcin E492 administered to HeLa cells lead to necrosis [13], alongside the previously mentioned caspase-1 activation, and the link between programmed necrosis and membrane pore formation [38], it is clear that microcin has a dual mechanism of apoptosis and pore-formation.

Whilst Pediocin PA-1 is much smaller in size compared to Microcin E492, they are both globular in shape as well as having residues of similar properties. Furthermore, the 3D modelling analysis revealed that they also bind to TLR4 in the same manner. Pediocin PA-1 was showed to have a greater cytotoxic effect against HeLa cells compared to HT29 cells, however the researchers were only able to speculate [11], if Pediocin PA-1 does interact with TLR-4 then one potential reason for this difference could be because TLR-4 is over-expressed in HeLa cells [39]. On the other hand, although TLR-4 is expressed in human colon cells, this appears to be mainly limited to crypt cells, alongside TLR-2 [40] (see Fig 9). Therefore, we would not expect as high levels of TLR-4 to be expressed in HT29 cells. This goes some way to explaining why we do not observe as extreme an effect in HT29 cells as we do in HeLa.

We did consider the possibility of Pediocin PA-1 acting on other receptors, however due COVID-19 restrictions the experimental data were not generated to further validate the hypothesis. It would be interesting to explore the effect Pediocin PA-1 may have on TLR-2 – which has been shown to have a major role in TLR2 recognition [42].

Divercin V41, like Pediocin PA-1, is produced from gram-positive bacteria; microcin however, is produced from gram-negative bacteria. Therefore, it would be interesting to explore how TLR-2 expression effects cytotoxicity, especially considering its important role in the recognition of gram-positive bacterial components. If Pediocin PA-1 does in fact interact with TLR-2, then this could go some way to explaining why Divercin, despite the high homology to Pediocin PA-1, does not display a cytotoxic effect to HT29 cells. Unfortunately, Divercin V41 has not been tested alongside HeLa (with high TLR-2/4 expression) and therefore it is difficult to compose more than a speculatory argument in regard to the mechanism. However, this observed difference in cytotoxicity despite high homology does give strong indication that Pediocin PA-1, like Microcin, has several cytotoxic effects which may be dose dependant. Going forward it would be interesting to carry out a comparative study on the cytotoxicity of Pediocin PA-1 and Divercin V41 in HeLa and HT29 cells of wild type, TLR-2 knockout and TLR-4 knockout.

The Enterocin AB heterodimer has also been shown to be an apoptotic inducer of HeLa and HT29, which means it must also interact with a TLR. Interestingly both homodimers of Enterocin induce apoptosis however this effect is enhanced when acting as a heterodimer [14]. The study did not use caspase inhibitors or identify the caspase that induced apoptosis, therefore it is not possible to comment on what TLR Enterocin AB acts on, or whether they act on separate TLRs each–hence the enhanced effect when working as a heterodimer. Pediocin PA-1 and Enterocin A have the highest homology of all the compounds assessed and are the most

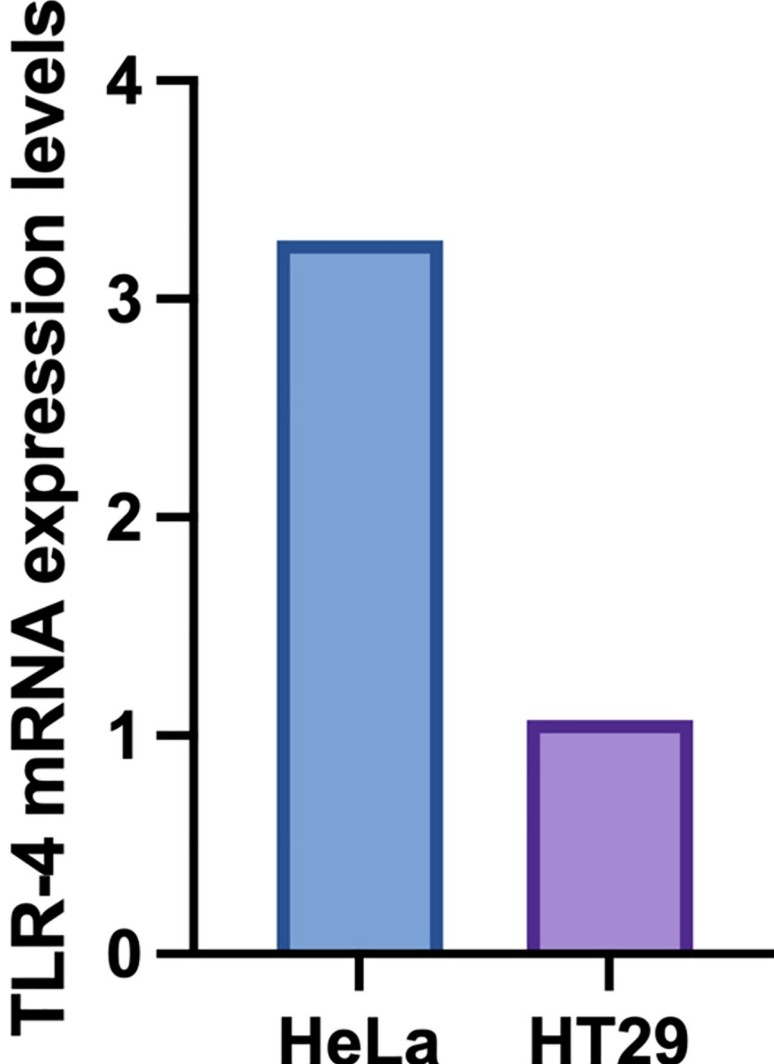

**Fig 9. mRNA expression level of TLR-4 in HeLa and HT29 of bioinformatic data obtained from the Broad Institute Cancer Cell Line Encyclopedia [41].**

similar in shape and structure. It is interesting then that inhibition of HeLa growth increased from 38.42% when Enterocin B only was being used, to 78.83% growth inhibition when Enterocin A was added [14]. Both Pediocin PA-1 and Enterocin A have cysteine residues, we know that in Pediocin PA-1 cystine residues lead to the formation of two disulphide bonds which stabilise the hairpin conformation of the two beta sheets. Therefore, we believe it is from studying Enterocin A which will give us the best clues about the mechanism of action of Pediocin PA-1. A possible experiment could be to use TLR knockout HeLa cells alongside Enterocin A to observe the effect of inhibition, fluorescent microscopy could also be used to visualise the toxin with the cell.

As well as evidence that Pediocin PA-1 induces apoptosis, it has been shown to target lipid vesicles as a dose dependant efflux of carboxyfluoerscein (CF). Imaging showed results were light scattering, meaning the lipid membrane was permeabilised but the overall structure was not changed [15]. Further support for this mechanism was seen when Pediocin PA-1 remained functional in the absence of protein receptors [43], from this it was concluded that Pediocin

did not interact with proteins, but rather was pore forming. We would argue that instead of this being Pediocin's only mechanism of cytotoxicity, it is one of at least two. The toxins in this research are all class II bacteriocins, class II bacteriocins have all been recognised for their pore-forming mechanisms in bacteria [44]. However, as highlighted above many have been shown to cause bioenergetic collapse secondary to apoptosis, Microcin as an example was shown to cause apoptosis however at high doses, necrosis was observed. This clearly shows class IIa bacteriocins appear to have a dose dependant cytotoxic effect.

We strongly feel that the findings of this study provide a strong argument and support for further, more targeted research into Pediocin PA-1 and other bacteriocins. With colon cancer expected to cause 52,980 deaths in the United States in 2021 [1], and cervical cancer behind the cause of two deaths a day in the UK [45], better treatment options need to be explored. With current conventional chemotherapy treatments causing significant central and peripheral neurotoxic effects [4], bacteriocins such as those studied here offer the ability to target cancer cells whilst avoiding damage to other healthy human cells. By understanding the mechanism by which Pediocin PA-1 works, we can either isolate it for use in further studies and trials or derive a synthetic compound which works in a similar way.

## Conclusion

Sequence alignment across bacteriocins studied in this project, including Pediocin PA-1, revealed mainly hydrophobic residues. The most common of these residues included: leucine, lysine and, glycine. It is also interesting to note that every bacteriocin also contained asparagine.

Visual Molecular Dynamic and SWISS-modelling revealed that bacteriocins which have previously shown a cytotoxic effect against HeLa and HT29 cells, were all globular in shape with polar surfaces. Whereas, Divercin V41, which has been shown to have no cytotoxic effect against HeLa cells, was predicted to be linear in shape. Considering that Divercin V41 was not dissimilar in sequence to the other bacteriocins studied, it could be suggested that a polar globular surface is an important factor for induction of apoptosis within tumour cells. This could be in terms of induction into the tumour cells, or specifically binding with TLR-4.

The computational work carried out in this study provides observations which support further *in vitro* studies to test the observations made in this study. Future experiments could include using TLR-4 knockout *HeLa* and *HT29* cells to observe if there is any change in the cytotoxic effect of Pediocin PA-1. If our conclusions of this study are true we would expect to see a significant reduction in effect of Pediocin PA-1 against these cancer cells. Western blot analysis could also be used to detect the presence of different caspases linked to other TLRs, in order to observe whether other TLRs are involved in the apoptosis of tumour cells. Dependant on the findings of these future studies, we could begin to try and develop a similar molecule to Pediocin PA-1, or mass production of Pediocin PA-1, for use in animal models of colon and cervical cancer. This would allow comparison of effectiveness and side effects against current conventional therapies. Building further upon the findings of this research could allow an improvement in the quality of life of cancer patients by reducing the harmful side effects of conventional therapies, as well as the development of a potentially more effective treatment.

## Supporting information

**S1 Fig. Sequence alignment of Microcin E492 and Pediocin PA-1.**
(TIF)

**S2 Fig. VMD 3D model of Pediocin and Microcin interacting with TLR-4.**
(TIF)

**S3 Fig. VMD 3D model of Pediocin PA-1 interacting with TLR-4.**
(TIF)

**S4 Fig. MultiAlign sequence alignment of Pediocin PA-1 and Enterocin heterodimer A and B.**
(TIF)

**S5 Fig. VMD 3D model of Enterocin A.**
(TIF)

**S6 Fig. SWISSMODEL 3D model of Enterocin B.**
(TIF)

**S7 Fig. MultiAlign sequence comparison of Divercin V41 and Pediocin PA-1.**
(TIF)

**S8 Fig. SWISSMODEL 3D Devised Model of Divercin V41 and Pediocin PA-1.**
(TIF)

**S9 Fig. mRNA expression level of TLR-4 in HeLa and HT29 of bioinformatic data obtained from the Broad Institute Cancer Cell Line Encyclopedia.**
(TIF)

## Author Contributions

**Conceptualization:** George P. Buss.

**Data curation:** George P. Buss.

**Formal analysis:** George P. Buss.

**Investigation:** George P. Buss.

**Methodology:** George P. Buss.

**Supervision:** Cornelia M. Wilson.

**Visualization:** George P. Buss.

**Writing – original draft:** George P. Buss.

**Writing – review & editing:** George P. Buss.

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
