## [Decision Letter · Decision Letter 0]

3 Jun 2021

PONE-D-21-14385

Exploring the cytotoxic mechanisms of Pediocin PA-1 towards HeLa and HT29 cells by comparison to known bacteriocins: Microcin E492, Enterocin Heterodimer and Divercin V41.

PLOS ONE

Dear Dr. Buss,

Thank you for submitting your manuscript to PLOS ONE. After careful consideration, we feel that it has merit but does not fully meet PLOS ONE’s publication criteria as it currently stands. Therefore, we invite you to submit a revised version of the manuscript that addresses the points raised during the review process.

We look forward to receiving your revised manuscript.

Kind regards,

Vivek Gupta

Academic Editor

PLOS ONE

Journal Requirements:

4. Please upload a copy of Figures 1-8, to which you refer in your text. If the figure is no longer to be included as part of the submission please remove all reference to it within the text.

Reviewers' comments:

Reviewer's Responses to Questions

**Comments to the Author**

1. Is the manuscript technically sound, and do the data support the conclusions?

Reviewer #1: Yes

Reviewer #2: Yes

2. Has the statistical analysis been performed appropriately and rigorously? 

Reviewer #1: N/A

Reviewer #2: N/A

3. Have the authors made all data underlying the findings in their manuscript fully available?

Reviewer #1: Yes

Reviewer #2: Yes

4. Is the manuscript presented in an intelligible fashion and written in standard English?

Reviewer #1: Yes

Reviewer #2: Yes

5. Review Comments to the Author

Reviewer #1: The qualities that make this research work more interesting are originality of the topic, technical quality, importance in medical field, overall representation and easy understandability, enough figures in line with the text written in the article.

This research work describes very nicely describes application of computational analysis using sequence alignment and modelling tools to investigate the mechanism of action of known bacteriocins.

The title represents manuscript's contents, the conclusions and interpretations are sound, the references are cited properly, this can be considered as unique research as this will help provide a fundamental understanding of cytotoxic mechanisms of Pediocin PA-1 towards HeLa and HT29 cells using molecular simulation tools.

Review and comments are as under -

I have a below recommendation for revisions to improve the readership of the manuscript –

1) Abstract

Line 33 – “Pediocin PA-1 and Microcin E492 and 3D modelling”, please replace “and” with “whereas”

Line 39- 42, Unfortunately, the COVID-19 pandemic meant that we were unable to carry out experiments in the lab, and the unavailability of important data meant we were unable to make solid conclusions but rather suggestions.

Please revise the statement as below –

Unfortunately, due to the COVID-19 pandemic, we were unable to carry out experiments in the lab, and the unavailability of important data we were unable to provide and validate our solid conclusions, but rather suggestions.

Line 42, Remove – “However”, however and despite mean the same. So, please use either one.

2) Introduction

Line 55, please remove “ ‘ “ after diagnosis

Line 56, replace “looks at” with “investigated”

Line 59, replace “ – (Hyphen)“ with a “ , (comma)”

Line 69, replace “looking into” with “discussing”

Line 71, replace “researched in greater detail” with “thoroughly researched or thoroughly evaluated.”

Line 73, remove “been”

Line 78 – 80, “against HT29 – being of the same class of toxin as Pediocin PA-78 1 we hope to identify what is different between these two bacteriocins, and as such gain greater insight into Pediocin PA-1’s mechanism of action.”

Please replace this statement with below –

“against HT29, despite it belonged to the same class of toxin as Pediocin PA-78 1 we hope to identify the difference between these two bacteriocins, and thereby gain greater insight into Pediocin PA-1’s mechanism of action.”

Line 82, please replace “when it has come” with “pertaining”

Line 83, please replace “the structure have been shown to form a disulphide bridge” with “the structure indicated a disulphide bridge formation”

Line 84, please replace “found” with “reported”

Line 86, please remove “been”

Line 86-87, please replace “permeabilization of the lipid bi-layer whilst also inducing

apoptosis by bio-energetic collapse” with “the lipid bi-layer permeabilization whilst inducing

apoptosis as well by bio-energetic collapse”

Line 88, please replace “been shown” with “demonstrated”

Line 90, please replace “has a” with exhibited”

Line 105, please replace “In this way we were able to give a strong argument for further study, as well”, with “In this way we were able to provide a fundamental understanding at molecular level for further investigation, as well”

3) Methodology –

For all the softwares used in this work such as UniProt, MultiAlin and VMD, please provide the version, name of developer, country of origin, etc.

Line 155, please replace “PBD” with “PDB”

Figure 1 is actually a table, so please convert this to a table and replace in-text citation pertaining to this table.

4) Results Microcin E492 –

Line 193, please replace “to the best of our knowledge, the best model” with “it was the only best possible model”

Line 196, please replace “come fold” with “come together to fold”

5) Results – Enterocin –

Line 227, please replace “research has highlighted that Enterocin A and B potentially have different mechanisms of action” with “research has highlighted potentially different mechanisms of actions for Enterocin A and B”

Line 228, please replace “by” with “upon” and “give” with “provide”

Line 240-241, please replace “there were no disulphide bridges detected” with “no disulphide bridges were detected”

6) Divercin V41 (Did you mean – “Results – Divercin V41”,please be consistent)

Line 275, please replace “could go someway to explain why Divercin V41 does not show” with “could explain up to certain extent, the reason for Divercin 41’s inability to”

Line 281, please replace “on analysis” with “upon investigation”

7) Discussion –

Line 308, please add a “comma” between that and when

Line 309, please add a “comma” between cells and necrosis

Line 315, please replace “-“ with “,”

Line 318, please remove “-“ and add a word “with” between alongside and TLR-2.

Line 323 – 324, please replace “limitations of experimental data available, and the COVID-19 restrictions which prevented us from carrying out lab work we were unable to effectively look at this.” With “COVID-19 restrictions the experimental data were not generated to further validate this hypothesis”

Line 341-344, this whole statement is unclear, please re-phrase it for better understanding.

8) Conclusion –

Please draw a conclusion. I was hoping to see a conclusion with a summary of current computational work. Also, state how this work can be extended to future similar work.

Reviewer #2: 1. Page 4, Line 85: Please provide details about the respective protein receptors here.

2. Page 4, Lines 84-91: These statements need further explanation. A clear reason behind the argument that “it is likely Pediocin PA-1 also has a dual mechanism of cytotoxicity” is missing here. Please elaborate.

3. Page 14, Lines 317-320: Authors are recommended to provide details or evidence on the TLR-4 over-expression in HeLa cells and limited expression in human colon cells.

4. Are there any Bacteriocins available for cancer therapy and how far they are successful?

5. It can be understood that authors were unable to carry out experiments in the lab due to the COVID-19 pandemic. However, how far can we rely on bioinformatic tools to while there is unavailability of important data? Please justify.

6. Conclusion needs be further clarified by the authors.

6. PLOS authors have the option to publish the peer review history of their article (what does this mean?). If published, this will include your full peer review and any attached files.

Reviewer #1: **Yes: **Harsh Shaiesh Shah

Reviewer #2: **Yes: **Vineela Parvathaneni

---

## [Author Response · Author response to Decision Letter 0]

8 Jul 2021

We find that all references are complete and correct - please see the rebuttal letter for the addition of two references. 

Please find all figures uploaded. 

The title has been amended on submission to match both manuscript and online submission.

We have checked the formatting and edited accordingly correcting: citation in brackets and heading formatting.

---

## [Editor Report · Decision Letter 1]

4 Aug 2021

PONE-D-21-14385R1

Exploring the cytotoxic mechanisms of Pediocin PA-1 towards HeLa and HT29 cells by comparison to known bacteriocins: Microcin E492, Enterocin Heterodimer and Divercin V41.

PLOS ONE

Dear Dr. Buss,

Thank you for submitting your manuscript to PLOS ONE. After careful consideration, we feel that it has merit but does not fully meet PLOS ONE’s publication criteria as it currently stands. Therefore, we invite you to submit a revised version of the manuscript that addresses the points raised during the review process.

While the authors have significantly amended the manuscript, potentially based on reviewers' comments, I do not see a detailed rebuttal letter addressing the comments. All I see is a generic letter with all edits summarized in a paragraph. Please review the comments from first revision, and provide detailed rebuttal when revising the manuscript.

We look forward to receiving your revised manuscript.

Kind regards,

Vivek Gupta

Academic Editor

PLOS ONE

Journal Requirements:

Additional Editor Comments (if provided):

While the authors have significantly amended the manuscript, potentially based on reviewers' comments, I do not see a detailed rebuttal letter addressing the comments. All I see is a generic letter with all edits summarized in a paragraph. Please review the comments from first revision, and provide detailed rebuttal when revising the manuscript.
---

## [Author Response · Author response to Decision Letter 1]

14 Aug 2021

Please find all corrections completed, and a separate letter entitled 'reviewer rebuttal'.

All available data for this study is contained with the manuscript as it was a bioinformatic study therefore supplementary data is not available/necessary.

---

## [Editor Report · Decision Letter 2]

18 Aug 2021

Exploring the cytotoxic mechanisms of Pediocin PA-1 towards HeLa and HT29 cells by comparison to known bacteriocins: Microcin E492, Enterocin Heterodimer and Divercin V41.

PONE-D-21-14385R2

Dear Dr. Buss,

We’re pleased to inform you that your manuscript has been judged scientifically suitable for publication and will be formally accepted for publication once it meets all outstanding technical requirements.

Kind regards,

Vivek Gupta

Academic Editor

PLOS ONE
---

## [Editor Report · Acceptance letter]

23 Aug 2021

PONE-D-21-14385R2 

Exploring the cytotoxic mechanisms of Pediocin PA-1 towards HeLa and HT29 cells by comparison to known bacteriocins: Microcin E492, Enterocin Heterodimer and Divercin V41. 

Dear Dr. Buss:

I'm pleased to inform you that your manuscript has been deemed suitable for publication in PLOS ONE. Congratulations! Your manuscript is now with our production department. 

Kind regards, 

on behalf of

Dr. Vivek Gupta 

Academic Editor

PLOS ONE